# BlockFFN: Towards End-Side Acceleration-Friendly Mixture-of-Experts with Chunk-Level Activation Sparsity

**Chenyang Song,**[*] **Weilin Zhao,**[*] **Xu Han,**[†] **Chaojun Xiao, Yingfa Chen,**
**Yuxuan Li, Zhiyuan Liu,**[†] **Maosong Sun**
Dept. of Comp. Sci. & Tech., Institute for AI, Tsinghua University, Beijing, China
{scy22,zwl23}@mails.tsinghua.edu.cn, {han-xu,liuzy}@tsinghua.edu.cn

## Abstract

To alleviate the computational burden of large language models (LLMs), architectures with activation sparsity, represented by mixture-of-experts (MoE), have attracted increasing attention. However, the non-differentiable and inflexible routing of vanilla MoE hurts model performance. Moreover, while each token activates only a few parameters, these sparsely-activated architectures exhibit low chunk-level sparsity, indicating that the union of multiple consecutive tokens activates a large ratio of parameters. Such a sparsity pattern is unfriendly for acceleration under low-resource conditions (e.g., end-side devices) and incompatible with mainstream acceleration techniques (e.g., speculative decoding). To address these challenges, we introduce a novel MoE architecture, BlockFFN, as well as its efficient training and deployment techniques. Specifically, we use a router integrating ReLU activation and RMSNorm for differentiable and flexible routing. Next, to promote both token-level sparsity (TLS) and chunk-level sparsity (CLS), CLS-aware training objectives are designed, making BlockFFN more acceleration-friendly. Finally, we implement efficient acceleration kernels, combining activation sparsity and speculative decoding for the first time. The experimental results demonstrate the superior performance of BlockFFN over other MoE baselines, achieving over 80% TLS and 70% 8-token CLS. Our kernels achieve up to $3.67\times$ speedup on real end-side devices than dense models. All codes and checkpoints are available publicly[1].

## 1 Introduction

To reduce the high costs of training and deploying large language models (LLMs), various efficient LLM architectures are proposed (Wan et al., 2023). A popular paradigm is designing architectures with **activation sparsity**, indicating that a considerable part of LLM parameters contribute weakly to LLM outputs given specific inputs, and thus can be skipped (i.e., not activated) in the forward and backward computation. Mixture-of-experts (MoE) is an outstanding representative and has been adopted by many recent models such as Mixtral-8×22B (Jiang et al., 2024) and DeepSeek-V3 (Liu et al., 2024b). Based on MoE, techniques including load balancing (Wang et al., 2024a) and expert parallelism (He et al., 2021) are adopted to achieve remarkable efficiency on cloud-side servers.

However, few efforts explore sparsely-activated architectures under low-resource conditions (e.g., end-side devices), where it is challenging to deploy huge MoE models and highly-distributed frameworks with expert parallelism. For end-side MoE models, which generally serve only a few users, some typical issues are not required to be considered (e.g., load balancing, see Appendix A), while raising the following two challenges.

**Performance compromise caused by imperfect routing.** Existing MoE models generally compromise performance due to two significant routing drawbacks: non-differentiability

---

[*]Equal Contributions.
[†]Corresponding Authors.

[1]https://github.com/thunlp/BlockFFN

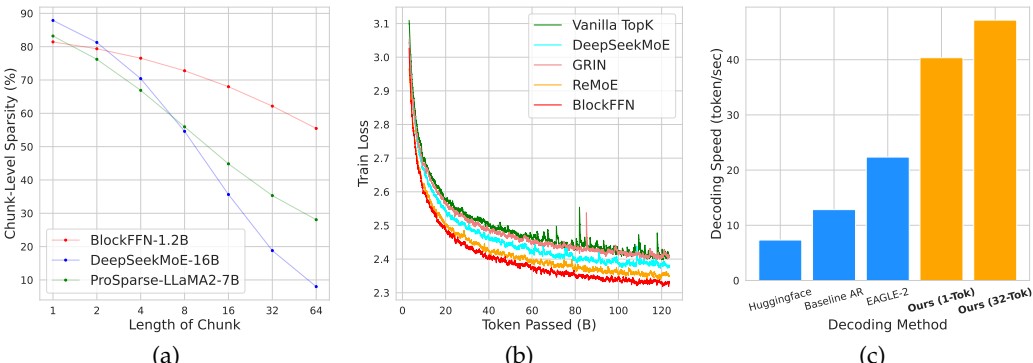

Figure 1: (a) For models with high TLS (except BlockFFN-1.2B), CLS quickly collapses to a lower level as a chunk contains more consecutive tokens. (b) Smoothed training curves of the BlockFFN-1.2B and other MoE baselines with the same total and activated parameters. (c) The speed of different decoding methods, where "Ours (1-Tok)" and "Ours (32-Tok)" are our token-level and chunk-level sparsity-based acceleration kernels, respectively.

and inflexibility. Specifically, most mainstream MoE models adopt a TopK router (Fedus et al., 2022; Jiang et al., 2024) with discrete and non-differentiable computation. Consequently, only activated parameters have complete gradients and are well updated at each step, which harms the convergence efficiency of MoE models (Liu et al., 2024c). Moreover, TopK makes each token activate the same number of experts, enforcing an inflexible activation pattern, which may weaken model performance (Huang et al., 2024). Few works can well alleviate both drawbacks, see Section 2.1.

**Acceleration unfriendliness caused by low chunk-level sparsity (CLS).** To make a sparsely-activated architecture more friendly for acceleration, just increasing the ratio of weakly-contributed experts for each token, namely the token-level sparsity (TLS), is not enough. Instead, the ratio of weakly-contributed experts for multiple consecutive tokens, namely the chunk-level sparsity (CLS), is critical for practical acceleration.

Specifically, a low CLS can eliminate the value of activation sparsity when combined with speculative decoding, a mainstream acceleration method that requires LLMs to process multiple consecutive tokens at the same time (Leviathan et al., 2023). Besides, other important resource-saving techniques, such as offloading, also become more challenging to implement due to the large differences in activation patterns within a specific chunk, leading to frequent GPU-CPU communication overheads. Unfortunately, existing works mainly focus on improving TLS (Mirzadeh et al., 2023; Song et al., 2024; 2025), but low CLS values still exist in most sparse architectures (see Figure 1a).

To address the above challenges, we introduce the following novel MoE architecture, as well as its training techniques and efficient end-side deployment.

For model architectures, we propose **BlockFFN**, a novel MoE paradigm that minimizes performance compromise. Specifically, its router module integrates ReLU and RMSNorm. ReLU computes differentiable and flexible activation patterns, enabling each token to determine the number of activated experts adaptively. RMSNorm generates learnable magnitudes of activation values. This separation of activation patterns and magnitudes alleviates the disturbance on activation magnitudes induced by regularization (e.g., the shrinkage of activation magnitudes caused by L1 (Rajamanoharan et al., 2024)), as regularization applies solely to the ReLU activation pattern. Through experiments, we demonstrate the better performance of BlockFFN compared to other MoE variants (Figure 1b, Table 2 and 3).

For training techniques, we introduce **CLS-aware training objectives** to improve the CLS of BlockFFN, which can make BlockFFN more friendly for acceleration, including the activation locality loss and the chunk sparsification loss. The former is to increase the similarity of activation patterns between neighbor tokens, helping reduce the gap between TLS and CLS. The latter is to increase the overall sparsity level. While existing sparsification objectives such as L1 (Song et al., 2025) are applied to each token independently, our chunk sparsification loss directly minimizes the probability that a specific expert is activated by

| Model | Activation values $\mathbf{A}(\mathbf{x})$ | Single expert outputs $E_i(\mathbf{x})$ |
|---|---|---|
| **Neuron-level activation sparsity** | | |
| Non-gated FFN | $\sigma(\mathbf{W}_{up}^T\mathbf{x})$ 
 $\mathbf{W}_{up} \in \mathbb{R}^{d_h \times N_e}$ | $\mathbf{w}_{down}$ 
 $\mathbf{w}_{down} \in \mathbb{R}^{d_h}$ |
| Gated FFN 
 (Shazeer, 2020) | $\sigma(\mathbf{W}_{gate}^T\mathbf{x})$ 
 $\mathbf{W}_{gate} \in \mathbb{R}^{d_h \times N_e}$ | $(\mathbf{w}_{up}^T\mathbf{x}) \cdot \mathbf{w}_{down}$ 
 $\mathbf{w}_{up}, \mathbf{w}_{down} \in \mathbb{R}^{d_h}$ |
| dReLU 
 (Song et al., 2024) | $\mathrm{ReLU}(\mathbf{W}_{gate}^T\mathbf{x})$ 
 $\mathbf{W}_{gate} \in \mathbb{R}^{d_h \times N_e}$ | $\mathrm{ReLU}(\mathbf{w}_{up}^T\mathbf{x}) \cdot \mathbf{w}_{down}$ 
 $\mathbf{w}_{up}, \mathbf{w}_{down} \in \mathbb{R}^{d_h}$ |
| **Block-level activation sparsity** | | |
| Vanilla TopK MoE 
 (Jiang et al., 2024) | $\mathrm{TopK}(\mathrm{Softmax}(\mathbf{W}_{router}^T\mathbf{x}))$ 
 $\mathbf{W}_{router} \in \mathbb{R}^{d_h \times N_e}$ | |
| DeepSeek-V1/V2 
 (Dai et al., 2024) | $[\mathbf{1}_{share}, \mathrm{TopK}(\mathrm{Softmax}(\mathbf{W}_{router}^T\mathbf{x}))]$ 
 $\mathbf{W}_{router} \in \mathbb{R}^{d_h \times (N_e - N_{share})}$ | |
| DeepSeek-V3 
 (Liu et al., 2024b) | $[\mathbf{1}_{share}, \mathrm{Norm}(\mathrm{TopK}(\mathrm{Sigmoid}(\mathbf{W}_{router}^T\mathbf{x})))]$ 
 $\mathbf{W}_{router} \in \mathbb{R}^{d_h \times (N_e - N_{share})}$ | $\mathbf{W}_{down}^T[\sigma(\mathbf{W}_{gate}^T\mathbf{x}) \odot \mathbf{W}_{up}^T\mathbf{x}]$ |
| DynamicMoE 
 (Huang et al., 2024) | $\mathrm{TopP}(\mathrm{Softmax}(\mathbf{W}_{router}^T\mathbf{x}))$ 
 $\mathbf{W}_{router} \in \mathbb{R}^{d_h \times N_e}$ | $\mathbf{W}_{gate}, \mathbf{W}_{up} \in \mathbb{R}^{d_h \times d_e}$ 
 $\mathbf{W}_{down} \in \mathbb{R}^{d_e \times d_h}$ |
| GRIN 
 (Liu et al., 2024c) | $\mathrm{SparseMixer}(\mathrm{Softmax}(\mathbf{W}_{router}^T\mathbf{x}))$ 
 $\mathbf{W}_{router} \in \mathbb{R}^{d_h \times N_e}$ | |
| ReMoE 
 (Wang et al., 2024b) | $\mathrm{ReLU}(\mathbf{W}_{router}^T\mathbf{x})$ 
 $\mathbf{W}_{router} \in \mathbb{R}^{d_h \times N_e}$ | |
| **BlockFFN** (Ours) | $\mathrm{RMSNorm}(\mathrm{ReLU}(\mathbf{W}_{router}^T\mathbf{x}))$ 
 $\mathbf{W}_{router} \in \mathbb{R}^{d_h \times N_e}$ | $\mathbf{W}_{down}^T\sigma(\mathbf{W}_{up}^T\mathbf{x})$ 
 $\mathbf{W}_{up} \in \mathbb{R}^{d_h \times d_e}, \mathbf{W}_{down} \in \mathbb{R}^{d_e \times d_h}$ |

Table 1: Comparison between different architectures with activation sparsity. $\sigma$ denotes an activation function (e.g., ReLU, Swish, GELU). $d_e$ is the intermediate dimension of each block-level expert. For simplicity, the expert index $i$ is omitted in notations of $E_i(\mathbf{x})$.

at least one token within the chunk. In experiments, we obtain average TLS values higher than 80% and 8-token CLS values higher than 70% (Table 2).

For end-side deployment, we implement **efficient acceleration kernels** for BlockFFN, combining activation sparsity and speculative decoding for the first time, and demonstrate its practical effectiveness on real end-side devices such as NVIDIA Jetson Orin NX. To enhance the efficiency of verifying multiple tokens in speculative sampling, we leverage the high activation similarity across multiple tokens induced by the high CLS level. This enables merging memory accesses to the same expert across different tokens, reducing the memory access volume to the union of experts activated by these tokens (Figure 2). Additionally, we implement the kernels based on CUTLASS (Thakkar et al., 2023) and utilize tensor cores to boost computational efficiency. Overall, the kernel achieves an acceleration ratio of $3.67\times$, compared to the baseline auto-regressive (AR) decoding (Figure 1c and Table 6).

## 2 Preliminaries and Related Works

### 2.1 Architectures with Activation Sparsity

To reduce the computation expenses of LLMs, various inference acceleration methods are proposed. Quantization (Xiao et al., 2023; Yao et al., 2023; Shao et al., 2023) and distillation (Gu et al., 2023; Hsieh et al., 2023) compress LLMs by using low bit-widths and transferring knowledge into smaller models, respectively. Weight pruning (Ma et al., 2023; Sun et al., 2023; Frantar & Alistarh, 2023; Xia et al., 2023) reduces FLOPs by removing weakly-contributed parameters (regardless of inputs). Speculative decoding (Li et al., 2024a; Cai et al., 2024; Zhao et al., 2024) uses a smaller model to generate multiple candidate tokens and lets the LLM itself verify these tokens in parallel.

Besides the above post-training methods, efficient architectures with activation sparsity can also effectively reduce the computation overhead of LLMs (Xue et al., 2024; Zhang et al., 2024b; Liu et al., 2024b). Specifically, in sparsely-activated architectures, a considerable part of the parameters contribute weakly to the model outputs given specific inputs.

In this work, we mainly focus on the activation sparsity within FFN layers. Typically, a sparsely-activated FFN with hidden dimension $d_h$ can be written in a unified MoE format:

$$\text{FFN}(\mathbf{x}) = \sum_{i=1}^{N_e} A_i(\mathbf{x}) \cdot E_i(\mathbf{x}), \quad \mathbf{A}(\mathbf{x}) = [A_1(\mathbf{x}), A_2(\mathbf{x}), ..., A_{N_e}(\mathbf{x})], \tag{1}$$

where $\mathbf{x} \in \mathbb{R}^{d_h}$ is the input hidden state, and $N_e$ is the number of experts. $A_i(\mathbf{x}) \in \mathbb{R}$ and $E_i(\mathbf{x}) \in \mathbb{R}^{d_h}$ denote the $i$-th activation value and expert outputs, respectively. If some $A_i(\mathbf{x})$ is zero or a low value, the corresponding expert $E_i$ is weakly-contributed. **Token-level sparsity** $TLS$ denotes the average ratio of weakly-contributed experts for a single token, while **chunk-level sparsity** $CLS_L$ is the ratio of experts contributing weakly to all tokens within a consecutive chunk of length $L$. Based on the granularity of experts, LLM architectures with activation sparsity can be divided into the following two categories.

**Neuron-level activation sparsity** commonly exists in mainstream LLMs (Li et al., 2022), where each expert is composed of a single neuron, i.e., single columns or rows within the FFN parameter matrices. For example, LLaMA2-7B is estimated to have about 70% TLS (Zhang et al., 2024a). However, as every single neuron is generally too small to be a memory access unit (i.e., < 1MB in BF16), most neuron-level sparse LLMs have bad memory locality. This makes it difficult to realize practical acceleration due to large IO overheads.

**Block-level activation sparsity** indicates that each expert is composed of multiple neurons or MLP modules. Represented by MoE (Fedus et al., 2022), such architectures are currently the mainstream solution for sparsity-based acceleration due to their better memory locality.

Nevertheless, the routing strategies of many MoE models, especially TopK routers, have non-differentiable and inflexible activation patterns, which limit model performance (Luo et al., 2024). Works such as GRIN (Liu et al., 2024c), ReMoE (Wang et al., 2024b), and DynamicMoE (Huang et al., 2024) try to address these issues. We list and compare several representative architectures of activation sparsity in Table 1. There are also special methods based on direct expert merging (e.g., SMEAR (Muqeeth et al., 2023) and Lory (Zhong et al., 2024)) that cannot be naively expressed as Equation 1.

In this work, BlockFFN absorbs the merits of both categories. With good memory locality of block-level experts, it adopts a ReLU-activated router (common in neuron-level settings), with activation values scaled by RMSNorm (i.e., the architectural difference from ReMoE).

## 2.2 Acceleration with Activation Sparsity

For neuron-level architectures, due to the relatively bad memory locality, designing tailored acceleration frameworks is complicated. Deja Vu (Liu et al., 2023) and PowerInfer (Song et al., 2023) utilize activation predictors to forecast the activation values, thus reducing IO overheads. PowerInfer-2 (Xue et al., 2024) introduces complex IO pipelines and neuron caches to promote higher speedup on specific smartphones. However, these all risk potentially inaccurate inference due to the imperfect performance of activation predictors.

Block-level architectures have relatively more available frameworks. FastMoE (He et al., 2021) and Tutel (Hwang et al., 2023) mainly focus on distributed training or inference with multiple GPUs working concurrently, while MegaBlocks (Gale et al., 2023) emphasizes the large-batch training of MoE. However, few of them are tailored for deploying MoE on end-side devices, where it is generally impractical to adopt a distributed implementation, and the service requirements shrink to small-batch inference for individual users. Under end-side conditions, sparsity-based acceleration will face different challenges.

As far as we know, **we present the first work to address the acceleration combining activation sparsity and speculative decoding**. Specifically, we improve the chunk-level sparsity of models through CLS-aware training objectives, making BlockFFN more friendly

for sparsity-based acceleration and speculative decoding. Moreover, our acceleration kernels are well applicable to end devices and have remarkable effectiveness.

## 3 Methodology

In this section, we first introduce the overall architecture of BlockFFN (Section 3.1) and CLS-aware training objectives (Section 3.2). Then, the acceleration kernels are introduced in Section 3.3, combining activation sparsity and speculative decoding for the first time.

### 3.1 BlockFFN Architecture

**Expert modules**   Considering the better memory locality of block-level activation sparsity, we make each BlockFFN expert an MLP with an activation function:

$$E_i(\mathbf{x}) = \mathbf{W}_{down}^{(i)T} \mathrm{Swish}(\mathbf{W}_{up}^{(i)T}\mathbf{x}), \tag{2}$$

where $i$ is the expert index, and $\mathbf{W}_{up}^{(i)} \in \mathbb{R}^{d_h \times d_e}, \mathbf{W}_{down}^{(i)} \in \mathbb{R}^{d_e \times d_h}$ are learnable weights.

Following DeepSeekMoE (Liu et al., 2024b), we use fine-grained expert segmentation to increase flexibility, namely $d_e << d_h$. We specifically add a Swish activation (Ramachandran et al., 2017) to increase the nonlinearity. Notably, we choose a vanilla non-gated MLP for experts instead of the more popular gated variant (Dauphin et al., 2017; Shazeer, 2020), as we find that a gated MLP can destroy the router sparsity (See Appendix I).

**Router module**   BlockFFN adopts a linear router with ReLU activation instead of TopK. As a common activation function in neuron-level sparse LLMs, ReLU is fully differentiable and can generate sparser activation patterns than other common activations (e.g., Swish) (Luo et al., 2024). Moreover, ReLU allows each token to adaptively activate different numbers of experts. This alleviates the inflexibility issue of conventional TopK routing.

On the other hand, as one major difference from ReMoE (Wang et al., 2024b), we add an RMSNorm layer (Zhang & Sennrich, 2019) after ReLU:

$$\mathbf{A}^0(\mathbf{x}) = \mathbf{W}_{router}^T\mathbf{x}, \quad \mathbf{A}^1(\mathbf{x}) = \mathrm{ReLU}(\mathbf{A}^0(\mathbf{x})), \quad \mathbf{A}(\mathbf{x}) = \mathrm{RMSNorm}(\mathbf{A}^1(\mathbf{x})), \tag{3}$$

where $\mathbf{W}_{router}$ is learnable parameters. Such a design makes the magnitude of activation values adaptively learned through RMSNorm, indicating better flexibility than vanilla softmax. Besides, RMSNorm separates the ReLU activation pattern $\mathbf{A}^1(\mathbf{x})$ from the final activation value $\mathbf{A}(\mathbf{x})$. This alleviates the disturbance on activation magnitudes by a direct regularization, which may hurt performance (Rajamanoharan et al., 2024) (Section 4.1.4).

### 3.2 CLS-Aware Training Objectives

The low chunk-level sparsity (CLS) is one important obstacle to fully leveraging activation sparsity in practical acceleration, especially under conditions where multiple consecutive tokens are processed in parallel (e.g., speculative decoding). The improvement of CLS involves two important aspects: (1) how to promote activation locality; (2) how to promote higher overall sparsity. We propose two respective training objectives.

**Activation locality loss**   Activation locality refers to the similarity of activation patterns between neighbor tokens, which also indicates the gap between TLS and CLS. To promote this property, we introduce the activation locality loss as an additional training objective:

$$\mathbf{A}_s^0(\mathbf{x}) = \mathrm{LeftShift}(\mathbf{A}^0(\mathbf{x})), \quad \mathcal{L}_{al} = \mathrm{BCE}[\sigma(\alpha \cdot \mathbf{A}^0(\mathbf{x})), \sigma(\alpha \cdot \mathbf{A}_s^0(\mathbf{x}))], \tag{4}$$

where $\sigma$ and $\alpha$ denote the sigmoid function and the sharpness hyper-parameter, respectively. We approximate the activation pattern through a sharp sigmoid function applied on $\mathbf{A}^0(\mathbf{x})$. LeftShift operator left-shifts a tensor in the sequence dimension, and finally, the binary cross entropy BCE minimizes the gap between the soft activation patterns of neighbor tokens.

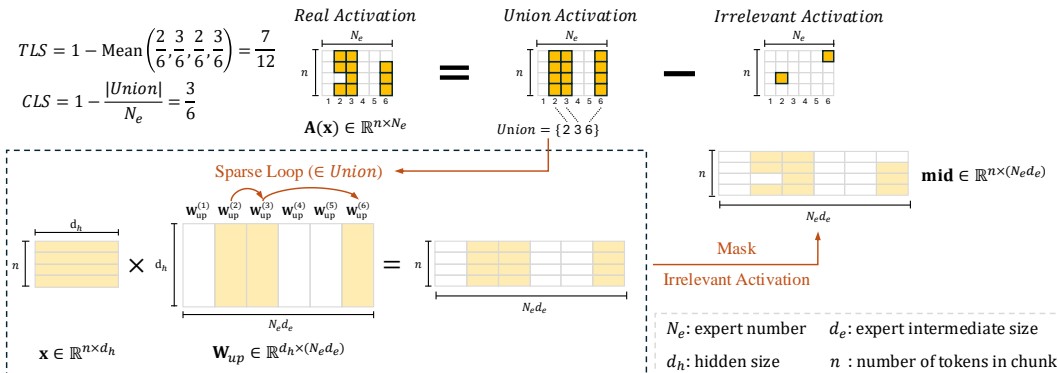

$$TLS = 1 - \text{Mean}\left(\frac{2}{6}, \frac{3}{6}, \frac{2}{6}, \frac{3}{6}\right) = \frac{7}{12}$$

$$CLS = 1 - \frac{|Union|}{N_e} = \frac{3}{6}$$

Figure 2: The overall framework of our acceleration kernels (the up projection part), which combines speculative decoding and chunk-level sparsity for higher efficiency. The down projection part has a similar implementation.

**Chunk sparsification loss** Despite the increase in activation locality, practical acceleration cannot be achieved without a considerable reduction in computation, which relies on a high sparsity level. Conventionally, L1 (Song et al., 2025) and router entropy (Huang et al., 2024) are both effective methods to improve sparsity, but they are applied independently to each token and cannot directly optimize the chunk-level sparsity.

Therefore, we design the chunk sparsification loss, which directly minimizes the chunk-level sparsity of a chunk with $L$ consecutive tokens. Suppose $p_{ik}$ is the probability of the $i$-th expert activated by the $k$-th token, while $\sum_{i=1}^{N_e} p_{ik} = 1$. The loss is the average probability that the $i$-th expert is activated by at least one token within this chunk (i.e., $\mathcal{P}_{act}^i$):

$$[p_{ik}]_{i=1}^{N_e} = \text{Norm}(\mathbf{A}^1(\mathbf{x})), \quad \mathcal{P}_{act}^i = 1 - \exp(\sum_{k=1}^{L} \ln(1 - p_{ik})), \quad \mathcal{L}_{cs} = \frac{1}{N_e} \sum_{i=1}^{N_e} \mathcal{P}_{act}^i, \quad (5)$$

where $\mathbf{A}^1(\mathbf{x}) \in \mathbb{R}^{N_e}$ specifically denotes the ReLU activation pattern of the $k$-th token, and Norm operator normailzes it in the expert dimension.

The overall training objectives are computed by: $\mathcal{L}_{total} = \mathcal{L}_{lm} + \lambda_{al}\mathcal{L}_{al} + \lambda_{cs}\mathcal{L}_{cs}$, where $\lambda_{al}$ and $\lambda_{cs}$ are corresponding factors. We introduce an **adaptive factor scheduler** to adaptively determine $\lambda_{cs}$ according to the dynamics of $\mathcal{L}_{cs}$, see Appendix B.

### 3.3 Acceleration Kernels

We implement acceleration kernels for BlockFFN, which are applicable to end-side devices and effectively combine chunk-level sparsity and speculative decoding.

Specifically, during the speculative sampling process, the draft model proposes $n$ draft tokens. When BlockFFN verifies these tokens, the router activation values are denoted as $\mathbf{A}(\mathbf{x}) \in \mathbb{R}^{n \times N_e}$, while the index union of activated experts for these $n$ tokens is $Union(\mathbf{x})$. Due to BlockFFN's high CLS level, the size of $Union(\mathbf{x})$ only accounts for a small ratio to the total expert number. Therefore, by only involving the experts in $Union(\mathbf{x})$ for computation, memory access is reduced and sparsity-based acceleration can be achieved in verification.

However, different experts may be activated by different subsets of tokens, which is not friendly for hardware parallelization. To address this issue, we leverage the characteristic that CLS and TLS values of BlockFFN are relatively close, indicating that each expert in $Union(\mathbf{x})$ is activated by the vast majority of tokens. Therefore, we only need to precompute all $n$ tokens for every activated expert for better GPU utilization, and subsequently discard computations induced by irrelevant activations. Specifically, for up projection, the hidden states of all $n$ tokens participate in the matrix multiplication with the experts in $Union(\mathbf{x})$, yielding an intermediate result **mid**, as illustrated in Figure 2. Finally, we apply a mask based on the sparse pattern of $\mathbf{A}(\mathbf{x})$ to remove irrelevant activations. Similar sparse computation is also conducted for down projection.

| Setting | Small | | | Medium | | | Large | | | XLarge | | |
|---|---|---|---|---|---|---|---|---|---|---|---|---|
| | $TLS$ | $CLS_8 \uparrow$ | PPL$\downarrow$ | $TLS$ | $CLS_8 \uparrow$ | PPL$\downarrow$ | $TLS$ | $CLS_8 \uparrow$ | PPL$\downarrow$ | $TLS$ | $CLS_8 \uparrow$ | PPL$\downarrow$ |
| Dense | - | - | 14.90 | - | - | 10.03 | - | - | 9.29 | - | - | 8.49 |
| TopK | 79.17 | 49.18 | 16.22 | 85.00 | 62.25 | 10.58 | 83.33 | 59.40 | 9.96 | 82.14 | 61.05 | 8.87 |
| DSMoE | 79.17 | 49.27 | 15.53 | 85.00 | 66.22 | 10.69 | 83.33 | 62.06 | 9.89 | 82.14 | 60.28 | 8.86 |
| GRIN | 79.17 | 50.45 | 15.50 | 85.00 | 61.48 | 10.40 | 83.33 | 59.08 | 9.72 | 82.14 | 60.89 | 9.03 |
| ReMoE | 78.33 | 42.44 | **14.60** | 84.43 | 52.00 | 10.42 | 82.80 | 50.79 | 9.60 | 81.93 | 51.01 | 8.78 |
| **BlockFFN** | 80.54 | **71.38** | 14.88 | 84.25 | **75.87** | **10.23** | 84.05 | **73.79** | **9.52** | 81.81 | **72.78** | **8.69** |

Table 2: The average perplexities (PPL) and chunk-level sparsity for 8 consecutive tokens ($CLS_8$) on the validation data under close TLS. "Dense" is the upper bound setting, which involves vanilla Transformers with the same parameter numbers as MoE settings.

| Setting | Small | | Medium | | Large | | XLarge | |
|---|---|---|---|---|---|---|---|---|
| | C.R.$\uparrow$ | R.C.$\uparrow$ | C.R.$\uparrow$ | R.C.$\uparrow$ | C.R.$\uparrow$ | R.C.$\uparrow$ | C.R.$\uparrow$ | R.C.$\uparrow$ |
| Dense | 45.15 | 32.72 | 52.68 | 44.85 | 54.89 | 48.87 | 57.30 | 49.53 |
| TopK | 43.73 | 28.99 | 50.90 | 42.47 | 52.84 | 43.05 | 54.98 | 49.29 |
| DSMoE | 44.35 | 31.90 | 50.46 | **44.89** | 52.54 | 46.45 | 55.28 | 50.57 |
| GRIN | 43.69 | 31.61 | 50.96 | 42.56 | 52.93 | 45.04 | 54.65 | 49.68 |
| ReMoE | **45.22** | 32.78 | 51.42 | 43.05 | 53.77 | 47.33 | 55.56 | 47.12 |
| **BlockFFN** | 44.80 | **33.93** | **51.75** | 43.74 | **54.44** | **51.60** | **56.42** | **50.73** |

Table 3: The average evaluation scores on two groups of benchmarks: commonsense reasoning (C.R.) and reading comprehension (R.C.). "Dense" is the upper bound setting.

The matrix multiplication kernel is modified based on CUTLASS GEMM (Thakkar et al., 2023), where we modify the outer loop of the up projection and the inner loop of the down projection to only scan through those activated experts in $Union(\mathbf{x})$, see Appendix J. To match the requirements of CUDA Tensor Core, we set the number of draft tokens $n$ to 32.

## 4 Experiments

### 4.1 Architecture Rationality

#### 4.1.1 Overall Results

To demonstrate the rationality of our architecture, we conduct experiments by comparing BlockFFN with multiple sparsely-activated architectures: Vanilla TopK MoE, DeepSeekMoE (DSMoE) (Dai et al., 2024), GRIN (Liu et al., 2024c), and ReMoE (Wang et al., 2024b) (see Appendix C). To ensure fairness, we keep consistent settings for attention layers and MoE experts (i.e., the number and intermediate dimension of experts) throughout baselines and BlockFFN. Besides, all settings (within each scale) have close parameter numbers, training token numbers, and token-level sparsity. We involve four parameter scales: Small (0.1B), Medium (0.5B), Large (0.8B), and XLarge (1.2B). See Appendix D for model settings.

We adopt two comparison metrics: perplexity (PPL) on validation datasets and evaluation scores on benchmarks. Benchmarks include two groups: commonsense reasoning (C.R.) and reading comprehension (R.C.). See Appendix E for details about data and benchmarks.

The PPL and evaluation scores are shown in Table 2 and 3, respectively. The training curves of the "XLarge" settings are drawn in Figure 1b. We can draw the following observations:

(1) *Performance*: Under close parameter numbers, all the settings (except for Small ReMoE and BlockFFN) cannot match the "Dense" setting, due to the performance compromise of sparsification. However, under close TLS values (i.e., identical average FLOPs for each token), **BlockFFN outperforms other MoE baselines in terms of validation PPL, train loss, and scores on downstream tasks**, showing less performance compromise.

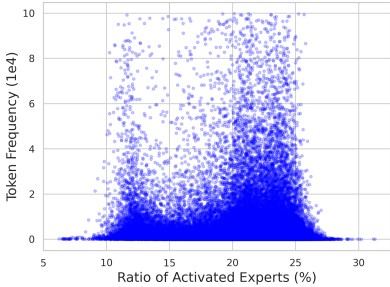
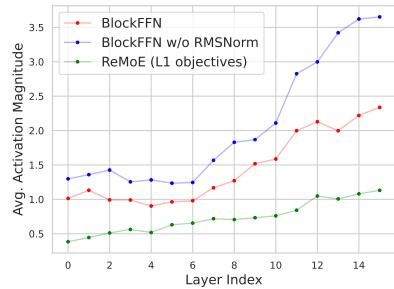

Figure 3: For each token in vocabulary, we calculate its frequencies and average ratios of activated experts, which show a bimodal distribution of expert allocation.

Figure 4: The layer-wise distributions of average activation magnitudes on the "Small" settings. While BlockFFN uses CLS-aware objectives, ReMoE adopts L1 regularization.

(2) *Sparsity*: Under close TLS values, **BlockFFN always has considerably higher CLS values than other baselines**. Attributed to CLS-oriented training objectives, this property makes BlockFFN more friendly for acceleration.

### 4.1.2 Expert Selection Stability

Low-resource conditions often require the implementation of memory-saving techniques, such as offloading, where the weights of experts are loaded into memory only when they are activated. Such a technique calls for higher expert selection stability. Specifically, the distribution of selected experts should be as similar as possible across consecutive tokens, so that the costs of expert IO can be saved.

In this section, we demonstrate that BlockFFN has significant expert selection stability, which is measured by the **reuse ratio**, namely, within the activated experts of one token, the average ratio of experts that are also activated by its next token. Within a sequence with $L > 1$ tokens, the set of experts activated by the $i$-th token is denoted by $\mathcal{S}_i$. The reuse ratio of this sequence is calculated by $\frac{1}{L-1} \sum_i^{L-1} \frac{|\mathcal{S}_i \cap \mathcal{S}_{i+1}|}{|\mathcal{S}_i|}$. As shown in Table 4, the high reuse ratios of BlockFFN models over 85% ensure satisfactory memory efficiency and good adaptability to offloading.

| Scale | Small | Medium | Large | XLarge |
|---|---|---|---|---|
| Reuse Ratio (%) | 90.28 | 89.69 | 87.13 | 89.57 |

Table 4: The average reuse ratios of BlockFFN models on the validation data.

### 4.1.3 Analysis of Expert Allocation

While ReLU-based routing is intrinsically differentiable, in this section, we examine how the router allocates experts and whether the inflexibility of activation patterns is truly addressed. Based on the validation data, we calculated the frequencies and average ratios of activated experts for each token, which are shown in Figure 3. These results demonstrate a **bimodal distribution of expert allocation**.

Specifically, the smaller peak lies between the activation ratio interval between 10% and 15%, which involves tokens such as numbers (e.g., "0", "1"), single characters (e.g., "a", "b"), and reserved words of programs (e.g., "import", "return"). These tokens have more deterministic meanings and thus require fewer experts for processing. By contrast, the larger peak between 20% and 25% mainly involves those tokens with more complex or diverse meanings, such as English pronouns and Chinese characters. Therefore, they need more experts to understand. Such a bimodal allocation of experts demonstrates that ReLU activation can truly address the routing inflexibility and allocate resources more wisely.

| Direct Ablation | | | | Substitute Objectives | | | |
|---|---|---|---|---|---|---|---|
| Setting | $TLS$ | $CLS_8 \uparrow$ | PPL$\downarrow$ | Setting | $TLS$ | $CLS_8 \uparrow$ | PPL$\downarrow$ |
| **AL+CS** | 80.54 | 71.38 | 14.88 | AL+L1 | 79.86 | 65.16 | 15.50 |
| CS | 81.67 | 67.56 | 15.66 | AL+Ent | 81.05 | 69.13 | 15.97 |
| AL | 63.55 | 52.59 | 14.89 | L1 | 79.35 | 45.13 | 15.01 |
| Null | 48.56 | 14.89 | 14.85 | Ent | 79.04 | 45.78 | 15.12 |

Table 5: Ablation studies on the training objectives. "**AL+CS**" is our standard setting. "AL", "CS", "L1" and "Ent" indicates activation locality, chunk sparsification, L1 norm, and router entropy, respectively. "Null" is the setting without any additional training objectives.

### 4.1.4 RMSNorm and Activation Magnitude Disturbance

In this section, we examine the effectiveness of the RMSNorm in our router module. First, we conduct an ablation study on the "Small" setting. **After removing the RMSNorm layer, the validation PPL rises from 14.88 to 15.04, indicating the effectiveness of RMSNorm.**

Moreover, to inspect the effects of RMSNorm, we calculate the average activation magnitudes (computed by L2 Norm) on the validation data. As shown in Figure 4, under all settings, higher layers (closer to output) generally have larger activation magnitudes. Without RMSNorm, the magnitudes of activation values in BlockFFN considerably rise with worse performance. By contrast, ReMoE, which has a similar architecture to BlockFFN without RMSNorm, suffers from significantly smaller activation magnitudes. This is attributed to the activation shrinkage issue induced by L1 regularization directly imposed on activation values (Rajamanoharan et al., 2024). We assume that RMSNorm, preventing activation values from direct regularization (Section 3.1), potentially alleviates activation magnitude disturbance and maintains a more stable and appropriate magnitude level.

Besides the above issues, expert granularity also has an important influence. Through experiments, we find that the validation loss generally decreases with finer experts, but the marginal benefits quickly diminish with > 40 experts for BlockFFN Medium. However, **the relationship between sparsity and expert granularity is nonmonotonic**, with the best setting of 40 experts achieving the highest sparsity. See Appendix F for more details.

## 4.2 Training Objective Rationality

In Section 3.2, we introduce **activation locality (AL) loss** and **chunk sparsification (CS) loss** as our training objectives. In this part, we mainly discuss whether such a practice is reasonable and better than other potential substitutes.

First, we conduct direct ablation studies by removing either AL loss or CS loss. As shown in the left part of Table 5, without AL (Setting "CS"), the model suffers from lower CLS and considerably higher PPL. On the other hand, without "CS" (Setting "AL" and "Null"), the sparsity (both $TLS$ and $CLS$) can be extremely low. These demonstrate the division of labor: **CS is mainly responsible for global sparsification, while AL is to promote the CLS with less performance compromise (compared with the direct application of a large CS loss)**. A possible explanation for why "AL+CS" performs better than pure "CS" is the competing relationship between "AL" and "CS". Specifically, the introduction of "AL" weakens the sparsification effect of "CS", producing lower TLS but higher CLS, and the better performance is attributed to the lower TLS level.

Next, we explore other potential substitute training objectives for sparsification. This includes the L1 norm (L1) (Song et al., 2025) and router entropy loss (Ent) (Huang et al., 2024). As shown in the right part of Table 5, replacing CS with L1/Ent (Setting "AL+L1" and "AL+Ent") can cause a considerable drop in performance. Besides, due to the absence of AL, "L1" and "Ent" cannot reach satisfactory CLS, either. Therefore, **CS is a more competitive sparsification partner of AL with less performance compromise**.

| Setting | MT. | Trans. | Summ. | QA | Math | RAG | Average |
|---|---|---|---|---|---|---|---|
| Huggingface | 7.61 | 7.32 | 7.54 | 7.81 | 7.76 | 5.96 | 7.33 (0.57×) |
| Baseline AR | 12.61 | 13.32 | 12.40 | 13.04 | 12.80 | 12.86 | 12.84 (1.00×) |
| EAGLE-2 | 23.70 | 20.26 | 20.74 | 21.99 | 25.01 | 22.47 | 22.36 (1.74×) |
| Ours (1-Tok) | 38.41 | **40.39** | 37.37 | 42.05 | 44.54 | 39.38 | 40.36 (3.14×) |
| **Ours (32-Tok)** | **49.43** | 39.18 | **42.70** | **46.09** | **59.85** | **45.76** | **47.17 (3.67×)** |

Table 6: Decoding speeds (token/sec) and average speedup ratios on NVIDIA Jetson Orin NX. "Ours (1-Tok)" is our token-level acceleration kernel purely dependent on sparsity, while "Ours (32-Tok)" is our efficient chunk-level acceleration kernels that combine EAGLE-2 and chunk-level sparsity. The speedup ratios are relative to "Baseline AR".

### 4.3 Practical Inference Acceleration

**Speedup experiment** To demonstrate the efficacy of our acceleration kernels, we conduct speedup experiments on Spec-Bench (Xia et al., 2024), a comprehensive benchmark for speculative decoding, with NVIDIA Jetson Orin NX 16GB. To ensure comparison fairness, except for the vanilla Huggingface auto-regressive decoding, all baseline methods are implemented within the framework of FR-Spec (Zhao et al., 2025), which applies CUDA kernels to reduce IO overheads and is much faster than Huggingface. These baselines include Baseline AR (i.e., a faster FR-Spec auto-regressive implementation), and EAGLE-2 (Li et al., 2024b). Moreover, since acceleration effects are more significant on larger models, we specifically train a 2.8B BlockFFN model as the base of our efficiency experiment.

From Table 6, we have the following observations: (1) "Baseline AR" is considerably faster than "Huggingface", indicating that our experimental framework is efficient enough and can alleviate the influence of potential IO overheads. (2) "Ours (32-Tok)", our acceleration kernel combining speculative decoding and activation sparsity, achieves the highest decoding speed, with 3.67× speedup. Meanwhile, it is faster than the pure sparsity setting "Ours (1-Tok)" and the pure speculative decoding "EAGLE-2". This demonstrates the value of such a combination and reveals the significant value of utilizing activation sparsity in end-side device inference acceleration. See Appendix H for speedup data on independent datasets.

**Upper bound analysis** Moreover, we conduct further experiments and find that both "Ours (1-Tok)" and "Ours (32-Tok)" can reach the theoretical upper bound of FFN speedup ratios induced by the corresponding token-level sparsity and the average union sparsity of tokens contained in an EAGLE-2 draft tree, see Appendix G.

## 5 Conclusion

In this work, we propose BlockFFN, a novel MoE architecture equipped with a ReLU-based differentiable and flexible routing strategy, which enables BlockFFN to outperform existing MoE counterparts. Next, we advocate more attention to chunk-level sparsity (CLS), and introduce the CLS-aware training objectives to promote the 8-token CLS to over 70%, offering BlockFFN better activation locality and more friendliness for end-side device acceleration. Finally, our efficient acceleration kernels achieve up to 3.67× speedup on NVIDIA Jetson Orin NX than the baseline auto-regressive decoding, reaching the sparsity-induced upper bound of FFN acceleration.

## Acknowledgments

This work is supported by the National Key R&D Program of China (No.2022ZD0116312), Beijing Municipal Science and Technology Plan Project (Z241100001324025) and a grant from the Guoqiang Institute, Tsinghua University. Our research is also supported by Huawei and can be carried out using the Huawei Ascend AI technology stack.

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

# A  Influence of Load Balancing

As a common practice of training MoE models, load balancing is to make the activation frequency of each expert as balanced as possible so that the model can be more friendly for the distributed deployment with expert parallelism, where experts are separately deployed on different devices and work concurrently. However, most end-side devices (which this work mainly focuses on) do not contain so many computation devices or cores, and thus cannot well support expert parallelism. Instead, it is more important to reduce global computation costs and promote activation locality, which is critical for end-side deployment techniques such as offloading and speculative decoding.

| Setting | $TLS$ | $CLS_8 \uparrow$ | PPL$\downarrow$ |
|---------|-------|-----------|------|
| **AL+CS** | 80.54 | 71.38 | 14.88 |
| **AL+CS+LB** | 78.06 | 69.68 | 15.26 |

Table 7: Load balancing is less important for end-side deployment and can cause potential performance degradation. "LB" indicates the load balancing with auxiliary loss.

Therefore, in this work, we do not consider load balancing and only focus on sparsification and the activation locality issue. Besides, load balancing can potentially cause performance degradation (Wang et al., 2024a). Specifically, as shown in Table 7, under similar TLS, the PPL suffers from a considerable increase after adding the load-balancing auxiliary loss.

# B  Adaptive Factor Scheduler for Chunk Sparsification Loss

With the language modeling loss $\mathcal{L}_{lm}$, the training objective is:

$$\mathcal{L}_{total} = \mathcal{L}_{lm} + \lambda_{al}\mathcal{L}_{al} + \lambda_{cs}\mathcal{L}_{cs}, \tag{6}$$

where $\lambda_{al}$ and $\lambda_{cs}$ are corresponding factors.

Considering the difficulty of tuning hyper-parameters, we introduce an adaptive factor scheduler for $\lambda_{cs}$, which controls the overall sparsity level. Concretely, this scheduler keeps $\lambda_{cs}$ constant as the initial value $\lambda_{cs}^0$ for the first $N_{st}$ steps. Next, for every $N_{adj}$ steps, the scheduler adjusts $\lambda_{cs}$ according to the change of $\mathcal{L}_{cs}$, increasing the factor when $\mathcal{L}_{cs}$ increases and vice versa. Formally, the behavior of this scheduler at step $m = (i+1)N_{adj}$ is:

$$\lambda_{cs}^{i+1} = \begin{cases} \lambda_{cs}^0 & \text{if } m \leq N_{st} \\ \gamma_{cs} \cdot \lambda_{cs}^i & \text{else if } \gamma_{cs} \leq 1 \\ \max(\gamma_{min}, \gamma_{cs}) \cdot \lambda_{cs}^i & \text{otherwise} \end{cases} \qquad \gamma_{cs} = \frac{\text{Avg}[\mathcal{L}_{cs}^t]_{t=i \cdot N_{adj}}^{(i+1)N_{adj}}}{\text{Avg}[\mathcal{L}_{cs}^t]_{t=(i-1)N_{adj}}^{i \cdot N_{adj}}}, \tag{7}$$

where $\mathcal{L}_{cs}^t$ denotes the loss value at step $t$, and $\gamma_{min}$ is the minimum magnification ratio.

# C  Experimental Settings

First, we give a detailed introduction to baseline MoE architectures used in our experiment:

(1) **Vanilla TopK MoE** is currently the most common MoE implementation, adopted by works such as Switch Transformer (Fedus et al., 2022) and Mixtral (Jiang et al., 2024). Their routers are composed of the softmax and TopK functions.

(2) **DeepSeekMoE (DSMoE)** (Dai et al., 2024) adopts a similar TopK MoE architecture but introduces shared experts for improvement, which are consistently activated by each token.

(3) **GRIN** (Liu et al., 2024c), also using the TopK activation, adopts an innovative routing strategy called SparseMixer-v2. This alleviates the non-differentiable issue through an approximation of the missing gradients.

(4) **ReMoE** (Wang et al., 2024b) makes the router differentiable by introducing a ReLU-based router module. Though similar to our design, ReMoE does not apply RMSNorm after ReLU,

| Scale | $d_h$ | $d_e$ | $N_e$ | $N_{layer}$ | $N_{tot}$ | $\lambda_{al}$ | $\lambda_{cs}^0$ | batch size | $n_{pre}$ |
|---|---|---|---|---|---|---|---|---|---|
| 0.1B (Small) | 768 | 64 | 48 | 16 | $1.19 \times 10^8$ | $2e-3$ | $5e-2$ | $1.57 \times 10^6$ | 10000 |
| 0.5B (Medium) | 1280 | 128 | 40 | 27 | $4.95 \times 10^8$ | $2e-3$ | $5e-2$ | $3.15 \times 10^6$ | 20000 |
| 0.8B (Large) | 1536 | 128 | 48 | 32 | $8.17 \times 10^8$ | $1e-3$ | $5e-2$ | $2.36 \times 10^6$ | 30000 |
| 1.2B (XLarge) | 1792 | 128 | 56 | 35 | $1.19 \times 10^9$ | $2e-3$ | $5e-2$ | $3.15 \times 10^6$ | 40000 |
| 2.8B | 2048 | 128 | 128 | 36 | $2.80 \times 10^9$ | $2e-3$ | $1e-1$ | $3.15 \times 10^6$ | 25000 |

Table 8: The major structural settings and hyper-parameters of our experimental models. $N_{layer}$, $N_{tot}$, and $n_{pre}$ denote the number of layers, the number of non-embedding parameters, and the pre-training steps, respectively.

and more importantly, its L1 regularization directly imposed on activation values can cause activation magnitude disturbance (Section 4.1.4) and harm performance.

Next, we list the hyper-parameters in Table 8. We adopt $\mu P$ parametrization (Yang et al., 2022) to promote training stability and reduce the influence of hyper-parameters. Therefore, we can adopt the same setting for the following parameters: peak learning rate $lr = 0.01$, $\beta_1 = 0.9$, $\beta_2 = 0.95$, weight decay $= 0.1$. We use the WSD scheduler to adjust the learning rates in the training process (Hu et al., 2024; Dubey et al., 2024). As for the adaptive factor scheduler, under all BlockFFN settings, we adjust the factor every $N_{adj} = 100$ steps, with $N_{st} = 1000$ and $\gamma_{min} = 1.025$.

# D Model Settings

For all settings, which include BlockFFN, the upper bound "Dense", and the other MoE baselines, we maintain the close number of total parameters, activated parameters (i.e., TLS), and training tokens. Moreover, the number and the intermediate dimension of experts are also exactly the same, following the fine-grained expert segmentation of DeepSeekMoE (Liu et al., 2024b). As for the attention layer, we apply the multi-latent attention (MLA) (Liu et al., 2024a) for models from 0.1B to 1.2B, while adopting group query attention (GQA) for BlockFFN-2.8B to make the acceleration implementation easier. Therefore, we ensure that the differences between different settings only lie in the routing strategy and training objectives, which are the key improved points of our work. The detailed structural settings of our models are listed in Table 8.

# E Datasets and Benchmarks

**Training data** The pre-training data of BlockFFN is a comprehensive mixture of multiple corpora across various categories. This includes C4 (Raffel et al., 2020), Pile (Gao et al., 2020), Dolma (Soldaini et al., 2024), CommonCrawl, StarCoder (Li et al., 2023), and other collected raw corpus. Besides, to obtain reasonable evaluation results, we perform a decay stage before evaluating models on benchmarks (Hu et al., 2024; Dubey et al., 2024). For this stage, instruction-tuning data are added, including EvolInstruct (Xu et al., 2023), UltraChat (Ding et al., 2023), OssInstruct (Wei et al., 2024), and other collected SFT datasets.

**Validation data** The validation data has the same distribution as the pre-training data. Deduplication is conducted to alleviate the intersections between pre-training and validation data, so that the validation data cannot be easily over-fitted.

**Evaluation benchmarks** The task-specific benchmarks used in our experiments can be divided into two groups: commonsense reasoning (C.R.) and reading comprehension (R.C.). The former group includes PIQA (Bisk et al., 2020), SIQA (Sap et al., 2019), and HellaSwag (Zellers et al., 2019). The latter group includes LAMBADA (Paperno et al., 2016), TyDi QA (Clark et al., 2020), and BoolQ (Clark et al., 2019). For both groups, the evaluation metric is 0-shot accuracy.

| Setting | Small | | | Medium | | | Large | | | XLarge | | |
|---|---|---|---|---|---|---|---|---|---|---|---|---|
| | PIQA | SIQA | Hella. | PIQA | SIQA | Hella. | PIQA | SIQA | Hella. | PIQA | SIQA | Hella. |
| Dense | 65.51 | 38.23 | 31.71 | 69.80 | 40.53 | 47.71 | 70.62 | 42.27 | 51.77 | 71.98 | 42.32 | 57.61 |
| TopK | 63.98 | 36.28 | 30.93 | 68.44 | 40.43 | 43.82 | 69.53 | 41.45 | 47.55 | 71.60 | 41.86 | 51.49 |
| DSMoE | 65.13 | 36.95 | 30.96 | 68.28 | 40.02 | 43.09 | 69.31 | 40.99 | 47.33 | 70.84 | 41.61 | 53.40 |
| GRIN | 64.20 | 36.08 | 30.78 | 68.28 | 40.02 | 44.59 | 69.31 | 41.35 | 48.13 | 69.86 | 41.76 | 52.34 |
| ReMoE | 64.91 | 38.18 | 32.56 | 69.31 | 40.38 | 44.57 | 71.38 | 40.38 | 49.55 | 71.06 | 42.22 | 53.40 |
| **BlockFFN** | 64.96 | 37.62 | 31.82 | 69.80 | 39.56 | 45.89 | 71.49 | 41.40 | 50.43 | 71.16 | 42.84 | 55.26 |

Table 9: The evaluation scores on the three benchmarks of commonsense reasoning (C.R.).

| Setting | Small | | | Medium | | | Large | | | XLarge | | |
|---|---|---|---|---|---|---|---|---|---|---|---|---|
| | LAM. | TyDi. | BoolQ | LAM. | TyDi. | BoolQ | LAM. | TyDi. | BoolQ | LAM. | TyDi. | BoolQ |
| Dense | 30.41 | 13.41 | 54.34 | 43.80 | 36.59 | 54.16 | 49.04 | 39.77 | 57.80 | 52.22 | 40.23 | 56.15 |
| TopK | 29.03 | 5.91 | 52.02 | 42.40 | 34.55 | 50.46 | 44.85 | 41.59 | 42.72 | 49.16 | 40.23 | 58.47 |
| DSMoE | 28.37 | 17.05 | 50.28 | 43.74 | 33.41 | 57.52 | 48.52 | 44.77 | 46.06 | 50.30 | 47.95 | 53.46 |
| GRIN | 28.41 | 12.27 | 54.16 | 42.77 | 35.00 | 49.91 | 44.87 | 38.18 | 52.08 | 50.44 | 44.55 | 54.04 |
| ReMoE | 29.44 | 15.68 | 53.21 | 42.58 | 32.05 | 54.53 | 45.90 | 39.55 | 56.54 | 51.35 | 30.23 | 59.79 |
| **BlockFFN** | 30.53 | 21.14 | 50.12 | 42.75 | 37.50 | 50.98 | 45.97 | 46.59 | 62.23 | 50.30 | 43.41 | 58.47 |

Table 10: The evaluation scores on the three benchmarks of reading comprehension (R.C.).

**Performance on independent benchmarks** In Table 3, we only list the average evaluation scores of two benchmark groups. In this section, we provide the evaluation results on independent benchmarks, as shown in Table 9 and 10.

# F Effect of Expert Granularity

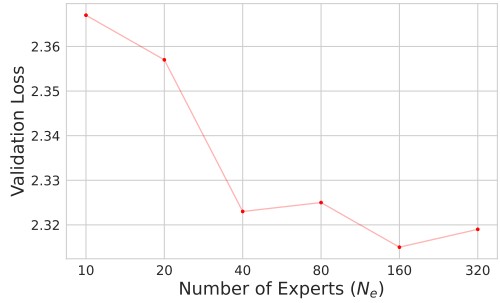

Figure 5: The validation loss of BlockFFN Medium with different expert granularities.

Figure 6: The TLS and CLS of BlockFFN Medium with different expert granularities.

Expert granularity has long been demonstrated to influence the performance of MoE models (Krajewski et al., 2024). Specifically, **given a fixed computation budget (assumed proportional to the parameter scale), what is the best trade-off between the expert number $N_e$ and expert dimension $d_e$?** To solve this problem, we conduct experiments on BlockFFN Medium with different expert granularities. These models are evaluated from four aspects: the validation loss, the token-level sparsity, the chunk-level sparsity, and memory locality.

First, as shown in Figure 5, while the loss drops considerably with coarse granularities (i.e., small $N_e$), the marginal benefits of granularity increase gradually diminishes with $> 40$ experts. Moreover, Figure 6 further displays a nonmonotonic relationship between sparsity and expert granularities. The setting with 40 experts, which we adopt in our main experiments (Section 4.1.1), achieves both the highest $TLS$ and $CLS_8$. Finally, as larger memory access units generally have better memory locality and hardware-friendliness, we

do not expect an extremely fine granularity. To sum up, 40 experts is the best setting for BlockFFN Medium. We leave more quantitative analyses for future studies.

## G  Upper Bound Analysis of Acceleration Kernels

| Setting | FFN Time (ms) | $TLS$ | Setting | FFN Time (ms) | $CLS_{spec}$ |
|---|---|---|---|---|---|
| Baseline AR | 61 | | EAGLE-2 | 88 | |
| **Ours (1-Tok)** | 7.8 (**12.8%**×) | $1 - \textbf{12.45\%}$ | **Ours (32-Tok)** | 27 (**30.7%**×) | $1 - \textbf{30.86\%}$ |

Table 11: The upper bound analysis of our kernels. $TLS$ and $CLS_{spec}$ values are evaluated on Spec-Bench decoding tokens, which are close to the FFN time consumption ratios of "Ours (1-Tok) / Baseline AR" and "Ours (32-Tok) / EAGLE-2", respectively.

To delve deep into the ability of our acceleration kernels, we conduct an upper bound analysis by inspecting the time consumption of FFNs separately. As shown in Table 11, "Baseline AR" and "EAGLE-2" can be viewed as the sparsity-ablation setting of "Ours (1-Tok)" and "Ours (32-Tok)", respectively, with only about 12.8% and 30.7% FFN time consumption. Surprisingly, we find that these two time consumption ratios are quite approximate to the $TLS$ and $CLS_{spec}$, respectively. Note that "Ours (32-Tok)" adopts a draft tree size of 32, and $CLS_{spec}$ is calculated by the average ratio of experts activated by the union of all the 32 tokens contained in the EAGLE-2 draft tree. This phenomenon indicates that both kernels can reach the theoretical speedup upper bound in FFN acceleration induced by the corresponding token-level sparsity and the union sparsity of tokens in a draft tree.

Notably, although our CLS-aware training objectives do not directly optimize the tree-level union sparsity, these objectives tailored for consecutive chunks are effective for tree patterns, since each path from the root node to the leaf node is still composed of consecutive tokens.

## H  Inference Acceleration on Independent Datasets

| Setting | MT. | | Trans. | | Summ. | | QA | |
|---|---|---|---|---|---|---|---|---|
| | Tokens/s | Speedup | Tokens/s | Speedup | Tokens/s | Speedup | Tokens/s | Speedup |
| Huggingface | 7.61 | 0.60 | 7.32 | 0.55 | 7.54 | 0.61 | 7.81 | 0.60 |
| Baseline AR | 12.61 | 1.00 | 13.32 | 1.00 | 12.40 | 1.00 | 13.04 | 1.00 |
| EAGLE-2 | 23.70 | 1.88 | 20.26 | 1.52 | 20.74 | 1.67 | 21.99 | 1.69 |
| **Ours (1-Tok)** | **38.41** | **3.05** | **40.39** | **3.03** | 37.37 | 3.01 | 42.05 | 3.22 |
| **Ours (32-Tok)** | **49.43** | **3.92** | 39.18 | 2.94 | **42.70** | **3.44** | **46.09** | **3.53** |

Table 12: Detailed speedup results on NVIDIA Jetson Orin NX (1st part).

| Setting | Math | | RAG | | Average | |
|---|---|---|---|---|---|---|
| | Tokens/s | Speedup | Tokens/s | Speedup | Tokens/s | Speedup |
| Huggingface | 7.76 | 0.61 | 5.96 | 0.46 | 7.33 | 0.57 |
| Baseline AR | 12.80 | 1.00 | 12.86 | 1.00 | 12.84 | 1.00 |
| EAGLE-2 | 25.01 | 1.95 | 22.47 | 1.75 | 22.36 | 1.74 |
| **Ours (1-Tok)** | 44.54 | 3.48 | 39.38 | 3.06 | 40.36 | 3.14 |
| **Ours (32-Tok)** | **59.85** | **4.68** | **45.76** | **3.56** | **47.17** | **3.67** |

Table 13: Detailed speedup results on NVIDIA Jetson Orin NX (2nd part).

In Table 6, we provide the decoding speeds on each dataset contained in Spec-Bench and the average speedup ratio. In this section, we list the speedup ratios on each independent dataset, as shown in Table 12 and 13.

| MT. | Trans. | Summ. | QA | Math | RAG | Average |
|------|--------|-------|------|------|------|---------|
| 2.73 | **2.17** | 2.38 | 2.67 | 2.83 | 2.57 | 2.66 |

Table 14: The acceptance lengths on each independent dataset of Spec-Bench.

On most datasets, "Ours (32-tok)" achieves the best inference efficiency. However, there exists an exception, "Translation" (**Trans.**), where "Ours (32-tok)" underperforms "Ours (1-tok)". This indicates that the combination of chunk-level activation sparsity and speculative decoding has worse performance than utilizing token-level activation sparsity alone. After careful examination, we find this is attributed to the shortest EAGLE-2 acceptance length on this dataset (Table 14), which hurts the efficiency of speculative decoding. Therefore, the sparsity-involved speculative decoding is more reasonable when speculative decoding works efficiently, generally with longer acceptance lengths and larger models.

## I  Ablation Studies on the Gated Expert Variant

For the design of BlockFFN expert modules, we choose the non-gated MLP instead of the more widely adopted gated variant. To support this choice, we conduct an ablation study on BlockFFN (Small). As shown in Table 15, using a gated MLP for expert modules can cause extremely low sparsity, which is quite a surprising result worth further study. A possible explanation may lie in the "competition" of sparsity between the router module and the expert module, indicating that the router sparsity and the expert sparsity vary in the opposite direction. Therefore, the higher sparsity of gated MLPs can significantly weaken the sparsity of the router module.

| Expert Design | $TLS \uparrow$ | $CLS_8 \uparrow$ | PPL$\downarrow$ |
|---------------|------|-------|-------|
| Non-gated | 80.54 | 71.38 | 14.88 |
| Gated | 28.39 | 25.44 | 15.17 |

Table 15: The ablation results of the gated variant for BlockFFN expert modules.

## J  More Details about Acceleration Kernels

Figure 7 shows more details about our efficient acceleration kernels. These include the execution details of the two-loop structure of the kernels, and how the weights of activated experts are transferred between different memory hardware (e.g., SRAM and HBM).

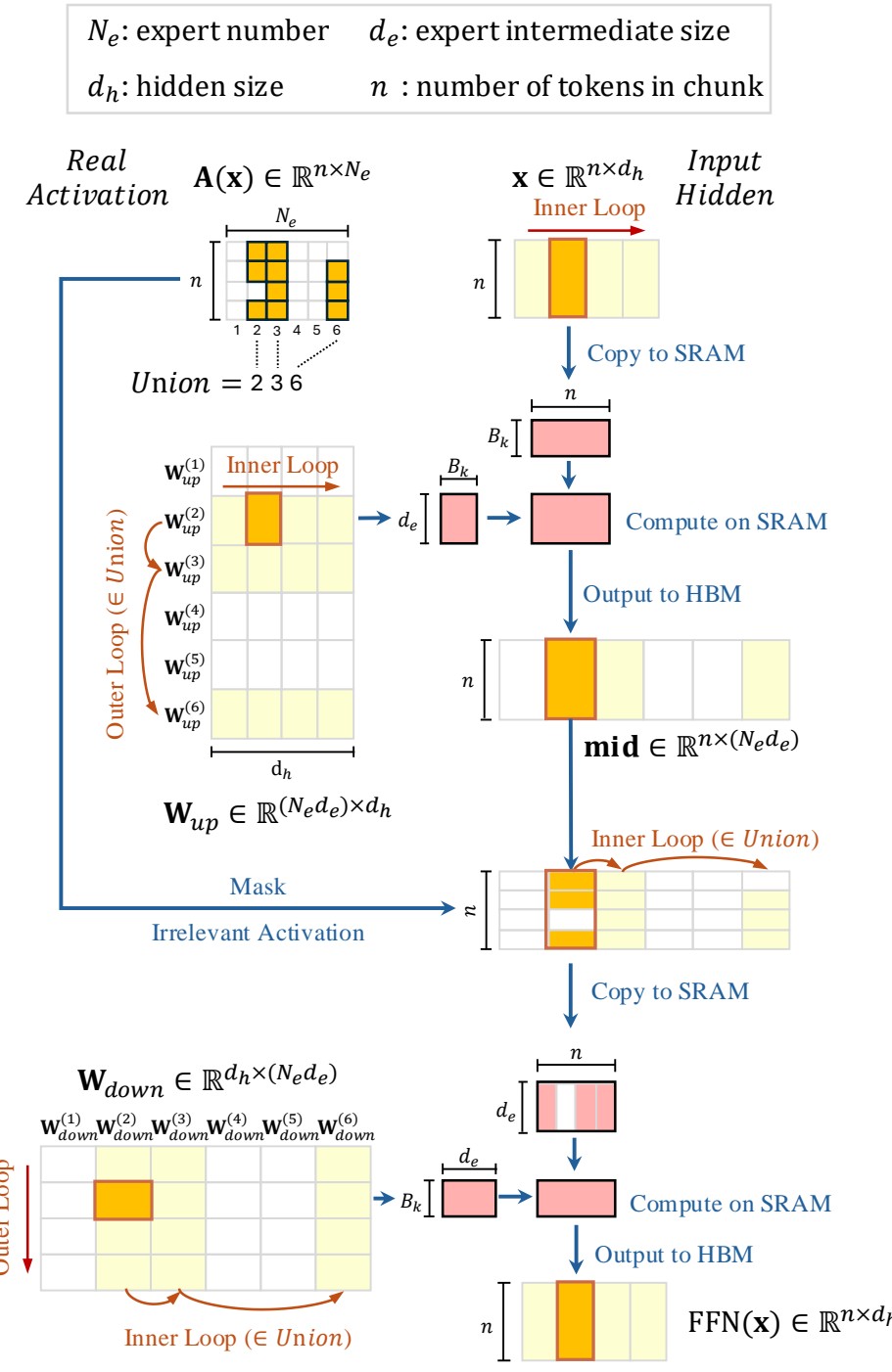

Figure 7: The detailed framework of our efficient acceleration kernels.

