# OpenReview forum: "BlockFFN: Towards End-Side Acceleration-Friendly Mixture-of-Experts with Chunk-Level Activation Sparsity"
_colmweb.org/COLM/2025/Conference — COLM 2025_

### Official Review · Reviewer_ohTG · 2025-05-13

**Rating:** 7
**Confidence:** 4
**Ethics Flag:** 1

**Summary:**

This work proposes a variant of mixture of experts for feed-forward layers with simpler sparsity in routing and chunk-aware loss. The router is inspired by ReMoE with additional root mean squared normalization to scale the activation. The activation locality loss is designed to differentiate with the activation pattern in neighboring position and chunk sparsification loss is designed to take into account the activations in prior history so that the expert routing is not skewed, but balanced for efficient computation. Experiments measured by perplexity and end-task performance, e.g., commonsense reasoning, shows clear gains when compared with other variants measured on models differentiating the number of parameters with faster inference on average.

**Questions To Authors:**

Please check the reason for rejection.

**Reasons To Accept:**

- The proposed mixture of experts, BlockFFN, is inspired by ReMoE in that the routing is normalized by root mean square to preserve invariants considering other activation norms. The feed-forward layer is further simplified by using the architecture from the original Transformer, not gated variants with scaling of up projection.
- The two loss functions for routing mechanism is inspired to trade the locality in neighboring position so that the activation pattern is clearly differentiated, while the chunk sparsitication loss considers locality when considering a history of activations. These losses are well motivated and ablation studies show gains.
- It is well engineered and descriptions are good enough to reproduce the results.
- Experiments are well designed, demonstrating the effectiveness of the proposed method.

**Reasons To Reject:**

- The motivation of BlockFFN is not clear and the ablation studies are not comprehensive. If my understanding is correct, each expert is basically identical to the feed-forward layer in the original Transformer, but others are using the gated variant. I'd like to see why the proposed expert is employing the simpler variant and what is the impact of using the simpler one for other routing methods, e.g., GRIN.
- The proposed loss functions are the major contribution of this work and they sound orthogonal to the routing method. I'd like to understand what is the impact when applying those losses to other routing methods.
- This work should run experiments to measure the impact of the token length in chunk sparsification loss (CLS), in order to quantify the gains of the approach. I suspect longer token length of $L$ will hurt the performance in, e.g., perplexity, but with the more efficiency in computation.

---

> ### Author Response · Authors · 2025-05-31
> **Rebuttal**
>
> Thank you for your excellent review! Your precious opinions encourage us to improve our work and forge ahead in the research path.
>
> In the following responses, "Weakness $i$" corresponds to the $i$-th point in "Reasons to Reject".
>
> # Weakness 1
>
> We use a simple non-gated FFN for experts for the following reasons:
>
> 1. **BlockFFN is originally designed as an MoE variant of a vanilla gated FFN**: In a vanilla dense gated FFN, as we know, it contains a gating projection, an up projection, and a down projection. In BlockFFN, we modify the gating projection into the ReLU-based router, and the up&down projections are widened and split into experts. This is why BlockFFN experts only have two linear projections without an extra "expert gating projection".
> 2. **For acceleration, simpler is better**: Non-gated experts, with only two linear projections, are potentially more friendly for the implementation of acceleration kernels, compared with gated experts with three projections.
> 3. **Gated experts help vanilla MoE, but not BlockFFN**: We provide the following ablation results, where NGE and GE indicate non-gated experts and gated experts, respectively.
>
> |         | BlockFFN+NGE | BlockFFN+GE | GRIN+NGE | GRIN+GE |
> | :-----: | :----------: | :---------: | :------: | :-----: |
> |   PPL   |  **14.88**   |    15.17    |  15.50   |  15.32  |
> |  $TLS$  |  **80.54**   |    28.39    |  79.17   |  79.17  |
> | $CLS_8$ |  **71.38**   |    25.44    |  50.45   |  50.10  |
>
> **For GRIN**, which has a fixed sparsity ratio, **gated experts can present better performance**. However, **for BlockFFN**, with ReLU-based dynamic activation sparsity, **gated experts do not bring about any improvement**. More severely, under the same training objectives, **gated experts can significantly eliminate the sparsity advantage of BlockFFN**.
>
> These findings indicate a fundamental difference in the properties of BlockFFN from many previous MoE methods. Admittedly, this is a strange phenomenon, and we do not find an explanation even months after COLM submission. This is also why we do not discuss the expert design in detail in the submitted article.
>
> Recently, we have surprisingly gained some insights from experiments. We find that in MoE with dynamic activation sparsity (e.g., BlockFFN and ReMoE), there exists **"sparsity competition" between experts and the router**. That is, the expert activation ratio and the router activation ratio vary in opposite directions. However, in MoE with fixed-sparsity routers (e.g., TopK and GRIN), there is no such competition.
>
> These findings may help explain some previous phenomena. For instance, the extremely low sparsity of "BlockFFN+GE" can be explained by sparsity competition. According to our experiments, gated FFN is easier to present high sparsity than non-gated FFN. Therefore, "BlockFFN+GE" has high expert sparsity, thus causing low router sparsity.
>
> # Weakness 2
>
> **The proposal of training objectives is highly dependent on the routing method.** In other words, ReLU-based dynamic routing is the basis of chunk sparsification and activation locality.
>
> On one hand, **chunk sparsification is just unsuitable for fixed-sparsity methods such as TopK and GRIN**. After all, what is the meaning of "sparsifying a fixed-sparsity model"? For these methods, token-level sparsity is a pre-defined value and cannot be adjusted by extra training objectives. The only possibility is ReMoE, but note that the RMSNorm and training objectives are just the two major improvements from ReMoE to BlockFFN.
>
> On the other hand, although the activation locality issue also exists in other MoE methods, **our activation locality loss cannot be trivially applied to baseline methods**. The key problem lies in how they determine activated experts. The activation locality loss is designed based on the property of ReLU routers, where experts with positive router scores are activated. However, most MoE settings just apply softmax/sigmoid to router scores and then choose activated experts based on the numerical order (i.e., sort & top-k). Without a zero activation threshold, it is just unreasonable to apply our activation locality loss.
>
> # Weakness 3
>
> Thank you for your suggestion. We provide an ablation study on $L$, the hyper-parameter in chunk sparsification loss. The validation perplexity values are shown in the following table.
>
> |   $L$   |  32   |  64   |  128  |  256  |
> | :-----: | :---: | :---: | :---: | :---: |
> |   PPL   | 15.41 | 14.88 | 16.04 | 15.93 |
> |  $TLS$  | 72.97 | 80.54 | 78.06 | 78.73 |
> | $CLS_8$ | 63.09 | 71.38 | 68.11 | 69.45 |
>
> The influence of $L$ on performance is nonmonotonic, with $L=64$ achieving the lowest PPL. By contrast, the impact of $L$ on chunk-level sparsity seems to follow a curve with decreasing marginal benefits. When we increase $L$ from 32 to 64, the sparsity significantly increases. However, with $L\leq64$, increasing $L$ may have little benefit in terms of efficiency improvement.

---

> > ### Comment · Reviewer_ohTG · 2025-06-07
> >
> > Thank you very much for the detail responses.
> >
> > * The empirical findings for the expert design sounds quite interesting and the additional insights into the trade of the sparsity in experts and routing seem to be the good explanation.
> > * Additional explanation for the loss function also resolves my concern.
> >
> > I'll adjust my scores according to the additional inputs from you.

---

> ### Comment · Area_Chair_t6Pn · 2025-06-07
> **Respond to rebuttal**
>
> Dear reviewer, could you please respond to the rebuttal?

---

### Official Review · Reviewer_G7oy · 2025-05-13

**Rating:** 7
**Confidence:** 4
**Ethics Flag:** 1

**Summary:**

This paper proposes BlockFFN, a novel Mixture-of-Experts (MoE) architecture designed for end-side device acceleration. The authors aim to improve both token-level sparsity (TLS) and chunk-level sparsity (CLS), and present an architecture that combines ReLU-based differentiable routing with RMSNorm to achieve this. The paper introduces CLS-aware training objectives and deploys a set of custom CUDA kernels to demonstrate the practical acceleration benefits on consumer-grade hardware (Jetson Orin NX).

The problem is timely and important: bringing MoEs to edge and desktop environments requires careful engineering and novel solutions. The authors show a comprehensive set of experiments that evaluate both hardware-level performance (decoding speeds) and language modeling metrics (perplexity, evaluation scores).

**Questions To Authors:**

- **Sparse TopK comparison**: In Table 5, your model shows strong speedups with single-token input, suggesting that activation sparsity is the primary source of improvement. Could a TopK MoE model with fewer-than-K activated experts (e.g., using thresholding) achieve similar results?

- **Memory implications of dynamic routing**: Have you analyzed whether expert selection is stable across token chunks? Could experts be preloaded per chunk and reused to reduce memory overhead?

- **RMSNorm vs ReMoE**: Is RMSNorm the only architectural difference between BlockFFN and ReMoE? If so, could you provide an ablation to confirm that it’s the primary contributor to performance improvements?

- **Server-side evaluation**: Since speculative decoding is more useful in server scenarios, have you evaluated BlockFFN on server-class GPUs (e.g., A100 or H100) with larger batch sizes? It would be helpful to understand how well your method scales in that context.

**Reasons To Accept:**

- **Timely and relevant**: Efficient MoEs for edge hardware are an important direction, and this work directly targets that gap.

- **Strong experimental results**: The model is thoroughly benchmarked both in terms of quality (perplexity, reasoning, QA tasks) and real-world decoding speed.

- **Hardware applicability**: Unlike many MoE papers, the authors test their system on actual consumer-grade hardware (Jetson NX), which gives weight to their claims.

- **Novel technical contributions**: The use of ReLU + RMSNorm for routing and the introduction of chunk-level sparsity training losses are fresh and well-motivated.

- **Batch size = 1 performance**: Table 5 shows substantial speedups even with single-token speculative decoding or no batching—this is especially important for local deployment, where batching isn’t practical.

- **Content-aware routing**: The observed bimodal expert activation distribution suggests that the routing policy is content-aware and semantically meaningful.

**Reasons To Reject:**

- **Speculative decoding is context-limited**: The paper includes speculative decoding as part of the proposed acceleration strategy, but this technique is primarily effective in server-side deployments where batch sizes >1 and longer contexts are common. On edge devices like Jetson Orin NX, where inference typically runs with batch size = 1, the gains from speculative decoding are less relevant.

- **Memory concerns with dynamic routing**: On small devices, loading different experts per token (especially when using ReLU) might force all experts to be kept in memory, which limits scalability. It’s unclear if expert selection is stable enough across tokens to allow chunk-wise caching or offloading.

- **Missing sparse TopK MoE baseline**: Table 5 shows that BlockFFN achieves significant speedups even for single-token inputs, which strongly suggests that activation sparsity itself, not speculative decoding, is the main driver of performance on the tested hardware. However, there is no comparison with a sparse TopK-based MoE that activates fewer experts per token (e.g., via adaptive thresholds). Without this, it’s unclear whether BlockFFN’s dynamic ReLU-based routing is essential.

- **Undiscussed training cost**: The paper doesn’t discuss the training cost of BlockFFN relative to baselines. The use of dynamic routing and auxiliary sparsity losses likely adds overhead that may not be trivial when scaling further.

- **Pretraining dataset scale unclear**: If BlockFFN is trained on a significantly larger or higher-quality dataset compared to the baselines, this could confound comparisons with other MoE variants. This needs to be controlled for.

- **ReMoE comparison could be better**: The authors state that RMSNorm is the main difference between BlockFFN and ReMoE, but no explicit ablation confirms this. A controlled test isolating the effect of RMSNorm would be helpful. This could be done by adding RMSNorm to ReMoE itself and doing the evaluation.

---

> ### Author Response · Authors · 2025-05-31
> **Rebuttal (2/2)**
>
> # Weakness 5
>
> We ensure that all settings (BlockFFN and baselines) are trained on **the same amount of tokens** with the **same data mixture recipe**. The experimental settings and datasets (Appendix C,E) hold for all settings, not just BlockFFN.
>
> # Weakness 6
>
> Thank you for your suggestion. First, the ablation of RMSNorm is already provided in Section 4.1.3 (line 264), which demonstrates that BlockFFN performs better than "BlockFFN w/o RMSNorm".
>
> Besides, we just experiment with "ReMoE+RMSNorm", and the validation perplexity values of different settings (0.5B) are shown in the following table. RMSNorm can indeed help ReMoE improve performance. The remaining gap between "ReMoE+RMS" and BlockFFN lies in different training objectives.
>
> | ReMoE | ReMoE+RMS | BlockFFN  |
> | :---: | :-------: | :-------: |
> | 10.42 |   10.34   | **10.23** |
>
> p.s. Actually, in recent experiments, we find replacing RMSNorm with MLP makes the performance even better, but at the cost of significantly lower sparsity. From the performance aspect, an important takeaway is just to make the activation magnitude independently decided by an extra module other than the router itself. A more complex extra module indicates better performance but lower sparsity. RMSNorm can provide a good balance.
>
> # Question 1
>
> See "Weakness 3".
>
> # Question 2
>
> See "Weakness 2".
>
> # Question 3
>
> Yes, the two major differences between ReMoE and BlockFFN are: (1) RMSNorm after router and (2) training objectives. We provide ablations in "Weakness 6".
>
> # Question 4
>
> Regretfully, since our target is to find **an efficient and well-performing LLM on edge devices**, we do not conduct experiments on server-class clusters (which is too expensive for us). In future research, we will design tailored methodologies for such server-class conditions.
>
> # References
>
> [1] Li, Yuhui, et al. "EAGLE-3: Scaling up inference acceleration of large language models via training-time test." *arXiv preprint arXiv:2503.01840* (2025).
>
> [2] Sadhukhan, Ranajoy, et al. "MagicDec: Breaking the latency-throughput tradeoff for long context generation with speculative decoding." *arXiv preprint arXiv:2408.11049* (2024).
>
> [3] Miao, Xupeng, et al. "Specinfer: Accelerating generative large language model serving with tree-based speculative inference and verification." *arXiv preprint arXiv:2305.09781* (2023).

---

> > ### Comment · Reviewer_G7oy · 2025-06-10
> > **Official Comment by Reviewer e3NV**
> >
> > **Official Comment by Reviewer G7oy**
> >
> > Thank you to the authors for the detailed responses. The additional experiments and clarifications were very helpful. Below I summarise how the rebuttal addresses my original concerns and where I still see room for improvement before the camera-ready.
> >
> > ## 1  Speculative decoding usefulness on edge devices  (resolved)
> > Your explanation—small-batch speculative decoding amortises *parameter-loading* rather than *compute*—is convincing, and the Jetson-NX results (Table 5) back it up. The 3.67 × end-to-end speed-up, with most of the gain coming from the sparsity-aware 32-token kernel, is a solid demonstration that SD is relevant even under `batch = 1`.
> >
> >
> >
> > ## 2  Expert reuse & memory footprint  (resolved, minor suggestion only)
> > The new *reuse-ratio* metric (~ 90 %) is very informative and suggests that a naïve per-token off-loading strategy would indeed benefit from high temporal locality. A brief discussion in the paper of how this translates into concrete GPU-RAM (or DDR/LPDDR) requirements would make the practical impact clearer, but I am satisfied that the routing is stable enough for chunk-wise caching.
> >
> >
> > ## 3  Necessity of ReLU + RMSNorm vs. sparse Top-K  (mostly resolved)
> > The rebuttal clarifies that at *matched activation ratio* BlockFFN out-performs Top-K, and that forcing Top-K to reach the same wall-clock speed would require dropping even more experts and hurt quality.
> > If time permits, a small *threshold-Top-K* baseline (e.g., variable *K* with λ-threshold) would be a neat quantitative confirmation, but I recognise this may be beyond the current revision cycle.
> >
> >
> > ## 4  Training cost and complexity  (minor, non-blocking)
> > I agree that, for edge-oriented models, inference efficiency matters more than training cost. Still, one sentence quantifying the *extra wall-clock time* introduced by the CS + AL losses (even if small) would help readers reproduce the method.
> >
> > ## 5  Dataset parity  (resolved)
> > Thank you for confirming identical token counts and mixture recipes across all baselines.
> >
> > ## 6  RMSNorm ablation vs. ReMoE  (resolved)
> > The new *ReMoE + RMSNorm* numbers (PPL 10.34 vs. 10.23) show that RMSNorm explains part, but not all, of the gap — supporting your claim that the CLS-aware objectives are also key.
> >
> > ### Additional positives from the rebuttal
> > * **Translation BLEU** on WMT20 adds evi*

---

> ### Author Response · Authors · 2025-05-31
> **Rebuttal (1/2)**
>
> Thank you for your excellent review! Your precious opinions encourage us to improve our work and forge ahead in the research path.
>
> In the following responses, "Weakness $i$" corresponds to the $i$-th point in "Reasons to Reject".
>
> # Weakness 1
>
> I do not agree with your statement that speculative decoding (SD) is less useful on edge devices with a small batch size. Actually, just as shown in the latest EAGLE-3 ([1] Table5), the throughput improvement of SD for a small batch is even more significant than for a large batch.
>
> The key issue lies in the different bottlenecks of these two situations. Under a small batch, the bottleneck lies in the parameter loading costs (while the computation burden is relatively small). The loading costs can be amortized by the parallel verification of consecutive tokens at the expense of more computation. For a large batch, decoding becomes computation-bounded, which by contrast makes verification costly. Thereby, in early works, SD for large batches is less encouraged [2,3].
>
> Of course, in SOTA frameworks, the challenges of large-batch SD are largely handled. We also admit the significant value of large-batch SD. However, we should not deny the rationality of small-batch SD. In summary, SD is very useful in both conditions, but with different challenges and techniques.
>
> # Weakness 2
>
> BlockFFN, especially the two training objectives, is just tailored for the edge devices with limited memory for the following reasons: (1) For each token, the number of experts that need to be loaded into memory is small thanks to the low token-level sparsity; (2) The high chunk-level sparsity is friendly for chunk-wise caching and offloading. For each chunk with 8 consecutive tokens, only no more than 30% of experts are activated on average. For 32 tokens, the chunk-level activation ratio is still lower than 40%.
>
> To better demonstrate the **expert selection stability**, we provide an extra metric **reuse ratio**: within the activated experts of a specific token, the average ratio of experts that are also activated by the next token. Suppose we use a simple offloading technique where we only reserve the activated experts when processing each token. The **reuse ratio indicates the ratio of experts that can be reserved in memory when we shift from one token to the next token**. The following results display a high near 90% reuse ratio for all BlockFFN scales. This well demonstrates the expert selection stability. Under offloading situations, most in-memory experts can be reused to reduce the memory overhead.
>
> | BlockFFN Scale  | 0.1B  | 0.5B  | 0.8B  | 1.2B  |
> | :-------------: | :---: | :---: | :---: | :---: |
> | Reuse Ratio (%) | 90.28 | 89.69 | 87.13 | 89.57 |
>
> # Weakness 3
>
> **The rationality of ReLU-based dynamic routing mainly comes from the performance concern**, not acceleration. Of course, a TopK MoE with a very low activation ratio can be quite faster, but at the expense of lower performance.
>
> As shown in Table2 and Table3, BlockFFN achieves better performance than TopK. This is measured at the same activation ratio, indicating that BlockFFN is better at the same FLOPS per token. Since our acceleration kernel (1-Tok) mainly utilizes sparsity, at the same activation ratio, BlockFFN and TopK MoE can run at close speeds, while TopK has worse performance.
>
> Therefore, if TopK MoE wants to achieve a faster speed (e.g., matching the BlockFFN 32-Tok acceleration kernel), it should activate even fewer experts and sacrifice more performance, causing a larger performance gap from BlockFFN. Such a fast but badly performing model is not what we want.
>
> # Weakness 4
>
> Indeed, the training cost is a very important issue. We do not think the auxiliary losses can cause much additional costs since their computational complexity is very low compared to the matrix multiplication operations of LLMs. However, we should admit that dynamic routing may cause trouble in training (e.g., the implementation of training frameworks may be more complex).
>
> Despite this fact, we want to emphasize that, the standpoint of our work is to find **an efficient and well-performing LLM on edge devices**. Considering the small parameter scale of edge LLMs, the inference efficiency is definitely more important than training. This is why we do not emphasize the training costs in the article.

---

> ### Comment · Area_Chair_t6Pn · 2025-06-07
> **Please respond to the rebuttal**
>
> Dear reviewer, could you please respond to the rebuttal?

---

> ### Author Response · Authors · 2025-06-10
> **Looking Forward to Response**
>
> We are looking forward to your responses, which are very important for us in improving our work.

---

### Official Review · Reviewer_SU7d · 2025-05-13

**Rating:** 7
**Confidence:** 3
**Ethics Flag:** 1

**Summary:**

This paper proposes a chunk-sparse MoE design that is better compatible with acceleration techniques. The results show improvement in block sparsity in routing decisions with equal or better PPL than other MoE methods. With the accompanying CUDA kernel and speculative decoding, the decoding speed is significantly higher. The paper is well-written, providing motivations and empirical evidence to substantiate the design decisions.

**Questions To Authors:**

1. I still find it hard to understand why CS loss hurts PPL while AL+CS helps. I think AL is mostly implied from CS. Improving the block sparsity naturally discourages abrupt changes in the routing distribution, except for the block boundaries. Could you provide more thoughts on why both are necessary? And why does CS alone cause so much performance degradation?

**Reasons To Accept:**

1. Realistic problem and solution: The paper tackles a previously overlooked realistic challenge, applying inference speed-up techniques for MoE models. Also, the authors didn't stop at "our method has higher block sparsity, making it possible for decoding speedup", but went all the way to develop the kernels to prove the speed gain.
2. Well-rounded experiments: This paper's experiments are extensive, providing sufficient evidence on both the model performance and speed-up. In addition, the method is supported by plenty of ablation and distribution analyses to help readers understand the proposed method and the effect of various components.

**Reasons To Reject:**

1. Remaining question on task performance: The performance evaluation focuses on language modeling, reading comprehension, and multiple-choice commonsense; it is still unclear whether BlockFFN can perform well on tasks that require long generation, such as the translation and summarization tasks from this paper's speed evaluation.
2. Complex method: The proposed method includes architecture design, loss regularizations, and particular scheduling for the introduced loss term that needs to be adopted together, rather than a single pluggable element. This increases the barrier to adoption and future extensions. It also makes it hard to find ways to convert a conventional pretrained model for the proposed method.

---

> ### Author Response · Authors · 2025-05-31
> **Rebuttal**
>
> Thank you for your excellent review! Your precious opinions encourage us to improve our work and forge ahead in the research path.
>
> In the following responses, "Weakness $i$" corresponds to the $i$-th point in "Reasons to Reject".
>
> # Weakness 1
>
> Thank you for pointing out the evaluation issue! Due to the time limit, we find it difficult to complete long-context training and evaluate our models on long-context tasks (e.g., $>8k$). Instead, we evaluate our models on the translation task of WMT20, as provided by UltraEval [1]. This benchmark includes both English-to-Chinese (en-zh) and Chinese-to-English (zh-en) translation. As demonstrated by the following evaluation results, BlockFFN can still achieve better translation performance (BLEU-4) than other baselines.
>
> |     Model     | WMT20-en-zh | WMT20-zh-en |
> | :-----------: | :---------: | :---------: |
> |   0.5B TopK   |   22.2654   |   11.9178   |
> |   0.5B GRIN   |   22.0765   |   11.9130   |
> |  0.5B ReMoE   |   22.0463   |   12.5296   |
> | 0.5B BlockFFN | **22.9080** | **12.8168** |
> |   0.8B TopK   |   24.5261   |   13.0567   |
> |   0.8B GRIN   |   24.0355   |   14.0157   |
> |  0.8B ReMoE   |   23.1456   |   13.5381   |
> | 0.8B BlockFFN | **25.0487** | **14.4182** |
> |   1.2B TopK   |   27.4604   |   15.2310   |
> |   1.2B GRIN   |   28.0665   |   15.0853   |
> |  1.2B ReMoE   |   27.8451   |   16.3561   |
> | 1.2B BlockFFN | **28.2929** | **16.6451** |
>
> # Weakness 2
>
> Thank you for your insightful opinion. Indeed, we admit that the current methodology is somewhat complex. We have already been working on the simplification. For example, we are experimenting with a joint training objective covering both the functions of "AL" and "CS". We will continuously improve the quality of our methodology.
>
> # Question 1
>
> Indeed, "AL+CS" performing better than "CS" is a surprising phenomenon found in our experiments. Although the paper only displays the ablation results on the 0.1B scale, this phenomenon has already been validated on multiple scales (e.g., 0.5B, 1.2B). We give the following explanation:
>
> 1. The "AL" loss is not a simple additional regularization to "CS". Instead, there exists a competing relationship between these two objectives. Specifically, the existence of "AL" can decrease the function of "CS".
> 2. "CS" is mainly responsible for sparsification. Therefore, "AL+CS" has less sparsification function, producing lower token-level sparsity and higher chunk-level sparsity.
> 3. According to experiments, the performance is negatively correlated with the token-level sparsity. Thereby, "AL+CS" has better performance than "CS" due to its lower token-level sparsity (i.e., higher activation ratio of parameters).
>
> Admittedly, the concurrent existence of two objectives brings about trouble in balancing them. Our ongoing work is already experimenting with a better joint training objective.
>
> # References
>
> [1] He, Chaoqun, et al. "UltraEval: A lightweight platform for flexible and comprehensive evaluation for llms." arXiv preprint arXiv:2404.07584 (2024).

---

> > ### Comment · Reviewer_SU7d · 2025-06-03
> >
> > Thanks for running experiments on translation tasks. My concern is resolved.

---

### Decision · Program_Chairs · 2025-07-08

**Decision:**

Accept

**Comment:**

This paper proposes a simple but effective approach for making MoEs more efficient by changing the activation function (ReLU followed by RMSNorm) as well as adding in additional objectives to encourage chunk-level sparsity (which is helpful for speculative decoding). The reviewers generally found the approach to be well-motivated, and the empirical results to be strong. Showing practical speedups on edge devices (section 4.3) was especially appreciated. On the negative side, there were some slight concerns with regard to baselines (e.g., comparison against a TopK baseline with a dynamic threshold), as well as whether this approach would work long-context tasks. However, most of these were addressed during the rebuttal, and I think this paper is a clear accept.